# Identification of Marker Compounds and In Vitro Toxicity Evaluation of Two Portuguese *Asphodelus* Leaf Extracts

**DOI:** 10.3390/molecules28052372

**Published:** 2023-03-04

**Authors:** Maryam Malmir, Katelene Lima, Sérgio Póvoas Camões, Vera Manageiro, Maria Paula Duarte, Joana Paiva Miranda, Rita Serrano, Isabel Moreira da Silva, Beatriz Silva Lima, Manuela Caniça, Olga Silva

**Affiliations:** 1Research Institute for Medicines (iMed.ULisboa), Faculty of Pharmacy, Universidade de Lisboa, 1649-003 Lisbon, Portugal; 2National Reference Laboratory of Antibiotic Resistances and Healthcare-Associated Infections, Department of Infectious Diseases, National Institute of Health Dr. Ricardo Jorge, 1649-016 Lisbon, Portugal; 3MEtRICs/Chemistry Department, Nova School of Science and Technology, Universidade Nova de Lisboa, 2829-516 Caparica, Portugal

**Keywords:** Aloe-emodin, antimicrobial, antioxidant, *Asphodelus*, chemical profile, herbal medicines, preclinical safety, *Staphylococcus epidermidis*

## Abstract

The leaves of *Asphodelus bento-rainhae* subsp. *bento-rainhae*, an endemic Portuguese species, and *Asphodelus macrocarpus* subsp. *macrocarpus* have been used as food, and traditionally as medicine, for treating ulcers, urinary tract, and inflammatory disorders. The present study aims to establish the phytochemical profile of the main secondary metabolites, together with the antimicrobial, antioxidant and toxicity assessments of both *Asphodelus* leaf 70% ethanol extracts. Phytochemical screenings were conducted by the TLC and LC-UV/DAD-ESI/MS chromatographic technique, and quantification of the leading chemical classes was performed by spectrophotometric methods. Liquid-liquid partitions of crude extracts were obtained using ethyl ether, ethyl acetate, and water. For in vitro evaluations of antimicrobial activity, the broth microdilution method, and for the antioxidant activity, the FRAP and DPPH methods were used. Genotoxicity and cytotoxicity were assessed by Ames and MTT tests, respectively. Twelve known compounds including neochlorogenic acid, chlorogenic acid, caffeic acid, isoorientin, *p*-coumaric acid, isovitexin, ferulic acid, luteolin, aloe-emodin, diosmetin, chrysophanol, and β-sitosterol were identified as the main marker compounds, and terpenoids and condensed tannins were found to be the major class of secondary metabolites of both medicinal plants. The ethyl ether fractions demonstrated the highest antibacterial activity against all the Gram-positive microorganisms, (MIC value of 62 to 1000 µg/mL), with aloe-emodin as one of the main marker compounds highly active against *Staphylococcus epidermidis* (MIC value of 0.8 to 1.6 µg/mL). Ethyl acetate fractions exhibited the highest antioxidant activity (IC_50_ of 800 to 1200 µg/mL, respectively). No cytotoxicity (up to 1000 µg/mL) or genotoxicity/mutagenicity (up to 5 mg/plate, with/without metabolic activation) were detected. The obtained results contribute to the knowledge of the value and safety of the studied species as herbal medicines.

## 1. Introduction

Medicinal plants have been used as potential functional foods or resources to prevent various diseases worldwide in different traditional medicine systems. Medicinal plants and their respective phytochemicals, mainly secondary metabolites, are used not only to combat specific nutrient deficiencies, but to sustain secure food and primary healthcare medicines [1].

The species *Asphodelus* L. (Asphodelaceae) is consumed in large quantities in the cuisines (e.g., soups, pastries, etc.) of several countries and cultures. The leaves of *Asphodelus aestivus* Brot., for instance, are commonly consumed as a cooked vegetable dish in Turkey, where they are known as “çiriş otu” [2]. In Puglia, on the southeast coast of Italy, burrata cheese is always wrapped in *Asphodelus ramosus* L. leaves to indicate the freshness of the cheese before it dries out [3]. In addition to their nutritional value, *Asphodelus* spp. leaves are widely used in traditional medicine to treat ulcers and urinary and inflammatory disorders [4]. In North African countries and the Iberian Peninsula, decoctions of leaves and stems have also been used to treat withering and paralysis [5,6]. Previously reported phytochemical studies of *Asphodelus* spp. extracts from leaves and aerial parts have revealed the presence of phenolic acids [7,8], flavonoids [6,7,8,9,10,11], and anthracene derivatives [8,12,13,14,15,16] as the main chemical classes of their marker secondary metabolites. Several in vitro and in vivo biological activities of *Asphodelus* spp. leaf and aerial parts extracts have been reported and documented for their antimicrobial [7,15,17,18,19,20], antioxidant [2,19,21,22,23], and antitumoral [7,15,21,24] activity [4].

*Asphodelus bento-rainhae* subsp. *bento-rainhae* P. Silva is an endemic species from Serra da Gardunha and is considered as “vulnerable” on the Red List of Threatened Species of the International Union for the Conservation of Nature (IUCN), and co-exists with *Asphodelus macrocarpus* subsp. *macrocarpus* Parl. in the same geographical area. They are known by the common Portuguese name “abrotea” (Ancient Greek: Ἀβρότονον), and their leaf is used as fertilizer and fodder in Portugal [25]. To date, no data related to the phytochemical characterization, pre-clinical safety, and biological potential of *Asphodelus bento-rainhae* and *Asphodelus macrocarpus* leaves have been found in the literature. Therefore, the present study was conducted to identify the main chemical constituents, antimicrobial and antioxidant activities of leaf extracts of these species along with their in vitro toxicity assessments, using samples collected at different times of the year to determine the most appropriate period for the collection of material and to contribute to the knowledge of safety and their value as herbal medicinal products.

## 2. Results and Discussion

### 2.1. Drug-Extract Ratio

The drug−70% ethanol extract ratio for *Asphodelus bento-rainhae* leaf (AbL) were 4.5: 1 and 4.8: 1 for the first (AbLa) and second (AbLb) collection seasons, respectively. For *Asphodelus macrocarpus* leaf (AmL), these values were obtained as 1:2.9 for the first (AmLa) and 1:6.3 for the second (AmLb) collection season.

### 2.2. Phytochemical Analysis

Thin-layer chromatography (TLC), followed by high-performance liquid chromatography (HPLC) coupled to a photodiode detector (UV/DAD), and electrospray ionization spectrometry (ESI/MS) techniques were applied for the rapid and reliable detection of several samples. The obtained chromatographic profiles of *Asphodelus bento-rainhae* and *Asphodelus macrocarpus* leaf extracts (AbLa, and AmLa, respectively) and their subsequent liquid-liquid partition with increasing polarity solvents, namely ethyl ether (AbLa-1, AmLa-1), ethyl acetate (AbLa-2, AmLa-2) and water (AbLa-3, AmLa-3), showed qualitative similarity in their chemical composition, characterized by the presence of terpenoids, phenolic acids, flavonoids, and anthracene derivatives. Based on both TLC and HPLC spectral analysis, using the authentic standards (co-chromatography) and comparison with literature data (Figure 1), twelve known compounds, namely, neochlorogenic acid (a), chlorogenic acid (b), caffeic acid (c), isoorientin (d), *p*-coumaric acid (e), isovitexin (f), ferulic acid (g), luteolin (h), aloe-emodin (i), diosmetin (j), chrysophanol (k), and β-sitosterol (l) were identified as major marker compounds of both species (Table 1, Figure 2).

Previously reported phytochemical studies of *Asphodelus* spp. revealed the presence of chlorogenic acid in the leaf and aerial part extracts of *Asphodelus aestivus* Brot. [8] and *Asphodelus ramosus* L. [26], while caffeic acid was only reported from the flower extract of *A. ramosus* [27].

Isoorientin from *Asphodelus aestivus* [8], *Asphodelus albus* Mill. subsp. *delphinensis* [10], *Asphodelus cerasifer* Gay [10,12], *Asphodelus microcarpus* Salz. et Viv. [6], and *Asphodelus ramosus* [11], together with isovitexin from *Asphodelus aestivus* [8] and luteolin from *Asphodelus acaulis* Desf. [12], *Asphodelus albus* [10,12], *Asphodelus cerasifer* [10,12], *Asphodelus fistulosus* L. [16], *Asphodelus macrocarpus* Parl. subsp. *rubescens* [12], *Asphodelus microcarpus* [7,10], *Asphodelus ramosus* [10], and *Asphodelus tenuifolius* Cav. [9] have also been recorded as the most common flavonoids of these species.

Aloe-emodin from *A. aestivus* [8], *A. albus* [12,13], *A. cerasifer* [12], *A. fistulosus* [14,16], *A. macrocarpus* subsp. *rubescens* [12], and *A. microcarpus* [13,14], as well as chrysophanol from *A. albus* [12,13], *A. fistulosus* [14,16], *A. macrocarpus* subsp. *rubescens* [12] and *A. microcarpus* [14,15] have been frequently detected and therefore found to be the most common anthracene derivatives.

*β*-sitosterol, a common phytosterol, was, however, only found in the root extracts of *A. albus*, *A. microcarpus*, and *A. tenuifolius* [6,28,29,30], and seed extract of *A. fistulosus* and *A. microcarpus* [31].

Quantification results of the main chemical classes of secondary metabolites, namely total phenolics (TPC), total flavonoids (TFC), total anthraquinones (TAC), total condensed and hydrolysable tannins (TCTC and THTC, respectively), together with total terpenoids, (TTC) are presented in Table 2.

Concerning the analysis between the different collection seasons for the same species, the results showed that the total content of TCTC and TFC in *A. bento-rainhae* (*p*-values: 0.034, 0.01, respectively) and THTC in *A. macrocarpus* leaf extracts were significantly higher in the first collection season (*p*-value: 0.01).

The analysis of the results between different species of the same collection season showed that TTC content in the first collection season and TFC content in the second collection season were significantly higher in *A. macrocarpus* when compared to those of *A. bento-rainhae* (*p*-values: 0.0021 and 0.01, respectively). However, TAC, TCTC, and TFC contents in the first collection season (*p*-value: 0.002, 0.007, and 0.006, respectively) and THTC content in the second collection season (*p*-value: 0.028) were significantly higher in *A. bento-rainhae* when compared to those of *A. macrocarpus*.

The obtained data showed the TCTC (180.96 ± 10.98 and 142.98 ± 6.71 mg CAE/g DW) and TTC (111.72 ± 22.77 and 165.47 ± 26.54 mg OAE/g DW) contents with the highest and TAC (1.07 ± 0.11 and 0.55 ± 0.07 mg RhE/g DW) with the lowest content in comparison to the other quantified chemical classes in both *A. bento-rainhae* and *A. macrocarpus* leaf extracts.

Previously reported *Asphodelus* spp. leaf extracts quantified values of TPC, TFC, and TCTC indicated the important role of solvent selection for the extraction procedure. In fact, *A. microcarpus* ethanol extract showed a higher amount of TPC and TFC (54.44 ± 13.6 mg GAE/g of DW and 31.13 ± 1.96 mg QUE/g of DW, respectively) in comparison to the aqueous and methanol extracts [21]. However, in *A. ramosus*, the aqueous extract exhibited a higher amount of TPC (33.51 ± 0.33 mg GAE/g of DW) when compared to the methanol, methanol/water (50%), and ethyl acetate extracts [32]. *A. aestivus* acetone extract also showed an elevated amount of TFC (17.74 ± 0.46 mg CAE/g of DW) in comparison to the aqueous, ethanol and methanol extracts [2,19]. Contrary to the data mentioned above and that obtained by us, significantly higher amounts of TPC (183.7 ± 3.5, 128.5 ± 2.1 and 109.7 ± 1.5 mg GAE/g of DW) and lower amounts of TCTC (59.8 ± 0.6, 49.2 ± 0.5 and 41.4 ± 0.3 mg CAE/g of DW) were reported from *A. tenuifolius* methanol, ethanol, and petroleum ether extracts [33]. It was also observed that both TPC and TFC contents have increased with the increase of the extraction temperature in the experiments done with *A. ramosus* [32].

### 2.3. Determination of In Vitro Antioxidant Potential

In this study, the antioxidant activity was evaluated by two complementary methods, DPPH assay to determine the 50% inhibition of free radical scavenging activity, and FRAP, which evaluates the reducing potency of the antioxidants to react to the ferric tripyridyltriazine (Fe^3+^-TPTZ) complex.

Concerning the results shown in Table 3, overall, *A. bento-rainhae* exhibited stronger antioxidant activity when compared to *A. macrocarpus* extracts. Among all the tested extracts, ethyl acetate fractions (AbLa-2, AmLa-2) showed the highest antioxidant activity when compared to all the other fractions (IC₅₀: 800 μg/mL and IC₅₀: 1200 μg/mL, respectively). When comparing FRAP and DPPH, the obtained an r value of −0.975, showing a strong correlation between them, validating the results of both techniques, although the data of the FRAP test correlate better with the quantifications data. The classes of compounds that correlate better with the antioxidant power of the extracts are the flavonoids (TFC, r value of 0.943) and phenolic compounds (TPC, r value of 0.949), in which a higher content of these compounds is related to higher antioxidant power. In accordance with these results, phytochemical screenings of the crude extracts and their L-L partitions revealed the presence of homoorientin and chlorogenic acid as the main marker compounds of most active fractions (AbLa-2, AmLa-2).

There is no report on the antioxidant activity of our studied *Asphodelus* species; however, the previously reported results of DPPH analyses of the other *Asphodelus* spp. showed that *A. microcarpus* leaf ethanol and methanol extracts exhibited the highest antioxidant activity (IC_50_: 55.9 μg/mL and IC_50_: 98 μg/mL, respectively) [21,23]. On the contrary, *A. aestivus* leaf methanol extracts noticeably showed a higher antioxidant activity when compared to ethanol extract (IC_50_: 160 μg/mL and IC_50_: 9540 μg/mL, respectively) [2,19]. *A. tenuifolius* leaf methanol extract exhibited the lowest IC_50_ (18370 μg/mL) levels among the others, including our studied species [22].

### 2.4. Assessment of the Antibacterial Potential

The in vitro quantitative method of susceptibility testing (determination of MIC values) was used for the evaluation of the antimicrobial potential against both selected Gram-positive and Gram-negative resistant pathogens in this study.

Concerning the obtained results, leaf crude extracts (AbLa, AmLa), and their subsequent ethyl acetate (AbLa-2, AmLa-2) and aqueous (AbLa-3, AmLa-3) L-L partition fractions did not exhibit antimicrobial activity against both Gram-positive and Gram-negative microorganism pathogens at any of the concentrations tested (MIC > 2000 µg/mL). However, as shown in Table 4, only diethyl ether fractions (AbLa-1, AmLa-1) demonstrated considerable antibacterial activity against all the Gram-positive microorganisms, with MIC values ranging from 62 to 1000 µg/mL. In general, *A. bento-rainhae* exhibited higher activity when compared to *A. macrocarpus,* and no activity in the tested range of concentrations (MIC > 2000 µg/mL) was found against Gram-negative microorganisms (*Escherichia coli*, *Klebsiella pneumoniae*, *Pseudomonas aeruginosa*, and *Acinetobacter baumannii*).

Aloe-emodin (compound i, Table 1), identified as one of the main marker compounds of the diethyl ether fraction of both plant extracts, was also tested against the pathogen panel under the study. This compound was found to be highly active against all the Gram-positive strains, particularly against all *Staphylococcus epidermidis* strains with a MIC between 0.8 to 1.6 µg/mL. In accordance with our results, aloe-emodin was previously reported as a potential antimicrobial that was active against several Gram-positive bacteria [34]; however, in a recent study, aloe-emodin with MIC values of 4 to 32 µg/mL exhibited deformities in the morphology of *S. epidermidis* cells and the destruction of the selective permeability of the cell membranes [35].

Results of studies involving the determination of the antimicrobial activity of other *Asphodelus* spp. against a similar pathogen panel revealed their lower antimicrobial potential. For instance, a leaf ethanol extract of *A. aestivus* exhibited a MIC of 42,000 µg/mL against *S. aureus*, and of 60,000 µg/mL against *Klebsiella pneumoniae* [36]. The *A. fistulosus* leaf ethanolic and aqueous extracts showed activity against *S. aureus* (MIC 2200 µg/mL and 7600 µg/mL, respectively) [37]. A methanol extract of *A. luteus* aerial part showed an MIC between 1250 to 2500 µg/mL against methicillin-resistant *S. aureus* (MRSA) [17]. A methylene-chloride extract of the aerial part of *A. tenuifolius* was found to be more active against *S. aureus* (MIC = 1600 µg/mL), *Enterococcus faecalis* (MIC = 1000 µg/mL), and *E. coli* (MIC = 1800 µg/mL) in comparison to the *n*-butanol and ethyl acetate extracts of the same species [9]. Recently, an *A. tenuifolius* whole plant chloroform extract was shown to be active against *S. epidermidis* (MIC = 580 µg/mL) [38]. *A. microcarpus* leaf extracts also showed antimicrobial activity against several Gram-positive strains, with MIC values of 78 to 5000 µg/mL [7,15,17,39,40]. *A. bento-rainhae* and *A. macrocarpus* leaf extracts seem to be more active against the tested Gram-positive strains in comparison to the other tested *Asphodelus* spp. extracts. The antibacterial activity of *A. fistulosus* leaf aqueous extract against *E. coli* (MIC = 62 µg/mL) and of *A. tenuifolius* aerial part methylene-chloride extract against the same microorganism (*E. coli*, MIC = 1800 µg/mL) and also against *P. aeruginosa* (MIC = 150 µg/mL) are examples of the few studies relating the antimicrobial activity of *Asphodelus* spp. to Gram-negative strains.

Overall, the observed antimicrobial activity of both *A. bento-rainhae* and *A. macrocarpus* leaf crude extracts were similar to those obtained and reported form the other *Asphodelus* spp. tested against a similar panel of pathogens. However, the fractionation of crude extracts enabled the detection of significant antimicrobial activity in the diethyl ether L-L partition fractions, quantitatively the richest in 1,8-dihydroxyanthracene derivatives, a known chemical class of secondary metabolites with antimicrobial activity [34].

### 2.5. Pre-Clinical Safety Assessment

Following the guidelines of the genotoxicity by the Ames test, which is commonly used as an initial screen of genotoxicity, for a substance to be considered genotoxic in the test, the number of revertant colonies on the plates containing the test compounds/substance must be more than twice the number of colonies produced on the solvent control plates (i.e., a ratio above 2.0). In addition, a positive dose-response should be evident for the various concentrations of the tested mutagen [41,42]. Since the crude extracts obtained from the first collection season (AbLa, AmLa) exhibited higher contents of the main classes of secondary metabolites, they were subsequently selected for further safety examination.

The obtained results of the Ames test for both AbLa and AmLa extracts are presented in Table 5. Neither extract induced an increase in the number of revertant colonies in any of the tested strains at any tested concentration, with (500, 1250, 2500, and 5000 µg/plate) and without (250, 625, 1250, 2500, 3750, and 5000 µg/plate) metabolic activation, when compared to the negative control. Moreover, cytotoxicity did not occur since there was neither a decrease in the number of spontaneous revertants nor a decrease on the background lawn of the plates at any of the concentrations tested. Therefore, under the conditions of this study, neither extract of the two species showed mutagenic activity.

Our cell viability assay (Figure 3) concurrently indicated that none of the AbLa and AmLa extracts reduced HepG2 viability. The AbLa extract (50–500 µg/mL) enhanced HepG2 viability/proliferation up to ~30% when compared to the 0 µg/mL concentration, whereas the same was observed for AmLa, i.e., it promoted HepG2 viability/proliferation by up to 40%, especially at higher concentrations (250–1000 µg/mL; *p* < 0.001 and *p* < 0.0001). Therefore, under the conditions of this study, the extracts of both species did not show mutagenic activity and in vitro cytotoxicity of HepG2, which is crucial to ensure their safety [42,43,44].

## 3. Materials and Methods

### 3.1. Chemical and Biological Reagents

Acetone, aluminum chloride, 2-aminoanthracene, 9-aminoacridine hydrochloride monohydrate, ammonium sodium phosphate dibasic tetrahydrate, ascorbic acid, benzo(*a*)pyrene, chlorogenic acid, chrysophanol, *d*-(+)-biotin, dimethyl sulfoxide/DMSO, 2,2- diphenyl-1-picrylhydrazyl/DPPH, gallic acid, glucose monohydrate, glucose-6-phosphate, diosmetin, neochlorogenic acid, nicotinamide adenine dinucleotide phosphate (NADP^+^), 2-nitrofluorene, *tert*-butyl hydroperoxide/T-BHP, 2,4,6-tris(2-pyridyl)-*s*-triazine/TPTZ and vanillin were obtained from Sigma-Aldrich (St. Louis, MO, USA). *p*-Anisaldehyde, ferric chloride hexahydrate, hydrochloric acid, l-histidine monohydrochloride monohydrate, magnesium acetate tetrahydrate, magnesium sulfate heptahydrate, methanol, perchloric acid, potassium iodate, sodium acetate trihydrate, sodium carbonate, sodium hydroxide, and sodium nitrite were purchased from Merck (Darmstadt, Germany). Aloe-emodin, caffeic acid, (+)-catechin, ferulic acid, isoorientin, isovitexin, luteolin, oleanolic acid, *p*-coumaric acid and rhein were acquired from Extrasynthese (Genay, France). Citric acid monohydrate, di-sodium hydrogen phosphate dihydrate, and sodium dihydrogen phosphate monohydrate were purchased from PanReac AppliChem (Barcelona, Spain). Sodium chloride and di-potassium hydrogen phosphate were from Honeywell Fluka™ (Seelze, Germany). β-sitosterol and 2-aminoethyl diphenylborinate were obtained from Acros organics (Geel, Belgium). Bacto™ agar was acquired from Becton Dickinson & Co (Franklin Lakes, NJ, USA), *n*-butanol came from Thermo Fisher Scientific^TM^ (Waltham, MA, USA), ethanol (CH_3_CH_2_OH) was sourced from Carlo Erba Reagents (Val-de-Reuil, França), ferrous sulfate heptahydrate came from M&B laboratory chemicals (Dagenham, UK), Folin-Ciocalteu was acquired from Biochem chemopharma (Cosne-Cours-sur-Loire, France), glacial acetic acid came from Chem-Lab NV (Zedelgem, Belgium), polyethylene glycol 400/PEG was sourced from VWR Chemicals (Rosny-sous-Bois, France), sulfuric acid (H₂SO₄) was acquired from PanReac AppliChem (Barcelona, Spain), sodium azide came from J.T. Baker Chemical Company (Phillipsburg, NJ, USA) and nutrient broth (NB) Nº 2 was sourced from Oxoid (Basingstoke, UK). Aroclor 1254-induced rat liver S9 was purchased from Trinova Biochem (GmbH, Giessen, Germany). In preparing all solutions, dilutions, and culture media, ultra-pure water from a Milli-Q water purification system, Millipore (Molsheim, France), was used.

### 3.2. Plant Materials

The leaves of *A. bento-rainhae* (AbL) and *A. macrocarpus* (AmL) were collected from Serra da Gardunha, Portugal, first at the early flowering stage (AbLa, AmLa) in Spring, and then for the second time (AbLb, AmLb), during the Summer of 2019. All samples were dried in a well-ventilated dark space at room temperature. Corresponding voucher specimens were deposited in the Laboratory of Pharmacognosy, Department of Pharmacy, Pharmacology and Health Technologies, Faculty of Pharmacy, Universidade de Lisboa (Voucher specimens’ number: OSilva_201901- *A. bento rainhae* and OSilva_201902- *A. macrocarpus*).

### 3.3. Preparation of Extract

Powder of the dried samples was obtained by grinding, and extraction was performed using the maceration method (with a mixture of ethanol/water 70:30) under agitation and filtration (3×, 24 h each). Hydroethanolic extracts were evaporated under reduced pressure at a temperature of less than 40 °C using a rotary evaporator and subsequently freeze-dried. The selected extracts (AbLa, AmLa) were then submitted to liquid-liquid partitioning (L-L), generating the diethyl ether (AbLa-1, AmLa-1), ethyl acetate (AbLa-2, AmLa-2), and aqueous (AbLa-3, AmLa-3) fractions.

### 3.4. Chromatographic Conditions

Silica gel 60 F_254_ and 60 RP-18 F_254_ pre-coated plates (Merck^®^, Darmstadt, Germany) were used for TLC screenings. Different spray reagents, including anisaldehyde–sulfuric acid for the detection of terpenoids, natural product polyethylene glycol reagent (NP/PEG = NEU) for the detection of phenolics, and potassium hydroxide (KOH) 5% ethanolic solution for the detection of anthracene derivatives [45] were used.

A HPLC-UV/DAD analysis was performed using a Waters Alliance 2690 Separations Module (Waters Corporation, Milford, MA, USA) coupled with a Waters 996 photodiode array detector (UV/DAD) (Waters Corporation, MA, USA). An Atlantis T3 column, RP-18 end-capped (5 µm, 150 × 4.6 mm), connected to a pre-column with the same stationary phase was used. The injection volume was 25 µL with a flow rate of 1 mL/min. A mixture of water + 0.1% formic acid (solvent A) and acetonitrile (solvent B) was used as the mobile phase, and gradients (95% A and 5% B), 20 min (71% A and 29% B), 30 min (67% A and 33% B), 35 min (64% A and 36% B), 45 min (50% A and 50% B), 65 min (100% B) and 75 min (95% A and 5% B) were applied. Crude extracts (20 mg/mL) were solubilized in water and standard solutions were prepared in acetonitrile (1 mg/mL) and filtered through a polytetrafluoroethylene syringe filter (0.2 µm). Data were collected and analyzed using a Waters Millennium^®^ 32 Chromatography Manager (Waters Corporation, Milford, MA, USA). The chromatogram was monitored and registered on *Maxplot* wavelength (240–650 nm).

An HPLC-MS/ESI analysis was carried out using an HPLC (Waters Alliance 2695), with an autosampler and photodiode array detector (Waters PDA 2996) in tandem with a triple quadrupole mass spectrometer (Micromass^®^ Quatro Micro^TM^ API, Waters^®^, Drinagh, Ireland) using an electrospray ionization source (ESI) operating in negative mode. A LiChrospher 100 RP-18 (5 µm) 250 × 4 mm column with respective pre-column (Merck, Darmstadt, Germany) was used. A mixture of water + 0.1% formic acid (solvent A) and acetonitrile (solvent B) was used as the mobile phase. Data were acquired and analyzed using MassLynx™ V4.1 software (Waters^®^, Drinagh, Ireland).

Peaks assignment and the identification of compounds were based on a co-chromatography technique with the comparison of retention times, UV-DAD, and mass spectral data with those of standards and published data.

### 3.5. Quantification Assays for Determination of the Main Classes of Secondary Metabolites

Total phenolic content (TPC) of the crude extracts was determined using the Folin-Ciocalteu method [46], and an increasing gallic acid calibration curve (10–70 µg/mL) was used to obtain the standard equation of Y = 0.0087X + 0.0264, R^2^ = 0.994. Total flavonoid content (TFC) was obtained following the method by Olivera et al., 2008 [47], and catechin concentrations (50–200 µg/mL) were used to obtain a standard curve with an equation of Y = 0.0039X + 0.027, R^2^ = 0.993. Total triterpenoid content (TTC) was assessed using the procedure developed by Chang & Lin, 2012 [48], and oleanolic acid concentrations (100–800 µg/mL in methanol) were used to obtain a standard curve with an equation of Y = 0.0012X + 0.0849, R^2^ = 0.994. For the determination of the total condensed tannins (TCTC) [46], catechin concentrations (200–2000 µg/mL) were used to obtain a standard curve with an equation of Y = 0.0002X + 0.0324, R^2^ = 0.981 and for quantification of total hydrolysable tannins (THTC) [49], gallic acid concentrations (100–600 µg/mL) was used to obtain a standard curve with an equation of Y = 0.001X + 0.054, R^2^ = 0.977. Total anthraquinones content (TAC) was evaluated according to the method described by Sakulpanich & Gritsanapan, 2008 [50], and rhein concentrations (3–18 µg/mL) were used to obtain a standard curve with an equation of Y = 0.0215X−0.0016, R^2^ = 0.998.

All of the above-mentioned colorimetric techniques were assessed in triplicate for method validation, and a UV-Vis spectrophotometer (Hitachi, U–2000, Tokyo, Japan) was used. Values were obtained using standard equations (where X was the concentration of standard equivalents expressed as milligrams per gram of dried extract and Y was the measured absorbance). All of the obtained data were treated statistically by a one-way analysis of variance (ANOVA) with the *Asphodelus* species as the source of variance. Once both of the *Asphodelus* species were collected in two different seasons, the obtained data were also analyzed by ANOVA, with the season as the source of variance. The significant value was set for a *p*-value < 0.05.

### 3.6. In Vitro Antimicrobial Activity

The antibacterial assay was carried out by the broth microdilution method [51] in 96-well tissue culture plates (VWR®, Radnor, PA, USA) to determine the activities by testing minimum inhibitory concentrations (MIC) of extracts against twelve reference (ATCC, LGC Standards S.L.U., Barcelona, Spain) and clinical strains (INSA clinical strains collection) of both Gram-positive (*Staphylococcus aureus*, *S. epidermidis*, *S. saprophyticus*, *S. haemolyticus*) (Table 6) and Gram-negative (*Escherichia coli*, *Klebsiella pneumoniae*, *Pseudomonas aeruginosa*, *Acinetobacter baumannii*) bacteria representing the antimicrobial resistance status. Samples to be tested were initially prepared in water or DMSO 10% and were screened at the concentration of 2–2000 μg/mL for crude extracts or L-L partitions and 0.2–200 for pure compounds. Serial dilutions were performed in a Mueller-Hinton medium and were distributed (50 μL) in each of the microplate wells using a microplate liquid handler (Precision^TM^ BioTek, Winooski, VT, USA).

For the preparation of inoculum from a pure bacterial culture on agar, a suspension in Mueller-Hinton medium (10^8^ CFU/mL) with a turbidity of 0.5 for Gram-negative and 0.25 for Gram-positive bacteria on the McFarland scale (Grant Bio™ DEN-1B, Cambridgeshire, UK) were prepared and stored at 4 °C until use. For MIC determination, the prepared suspensions were diluted at a ratio of 1:10, and from this dilution, 50 µL was added to all the wells. Two controls were included for each extract, fraction or compound, one plate in the absence of the extract solution and the other in the presence of the solvent (DMSO), to verify the absence of contamination and to check the validity of the inoculum. After incubation at 37 °C for 18 h, the plates were read in a lighted place, and the MIC was determined. All experiments were carried out in triplicate, as previously described, to obtain consistent values.

### 3.7. In Vitro Antioxidant Activity

The antioxidant potential was determined by two methods, initially started by a modified free radical scavenging activity (DPPH method) [52], followed by the ferric reducing antioxidant power test (FRAP assay). DPPH solution (3.9 mL, 6 × 10^−5^ M in methanol) was mixed with 100 µL of diluted extracts or standard (ascorbic acid). After 30 min of incubation at room temperature, the absorbance of samples and standard solutions was measured at 517 nm. The percentage of DPPH free radical scavenging activity was calculated using the following formula: % scavenging = [absorbance of control−absorbance of test sample/absorbance of control] × 100. Results were expressed as mean ± standard deviation and presented in inhibitory concentration (IC_50_ value), representing the sample concentration required to scavenge 50% of the DPPH free radicals.

For the Frap assay [53], 100 µL of plant extracts (1000–5000 μg) were mixed with 3 mL of working FRAP reagent (300 mM acetate buffer pH 3.6, 10 mM TPTZ in 40 mM HCl and 20 mM FeCl_3_. 6H_2_O in the ratio of 10:1:1 at the time of use); thereafter, samples were placed in the water bath at 37 °C. The reduction of ferric tripyridyl triazine (Fe III TPTZ) complex to ferrous form (which has an intense blue color) can be monitored by measuring the change in absorption at 593 nm, measured after 4 min. Ascorbic acid concentrations (25–175 µg/mL) were used to obtain a standard curve with an equation of Y = 0.616X−1.1702, R^2^ = 0.9989. The FRAP reagent was used as a blank, and results were expressed as mmol ascorbic acid/g dry extract. Values were obtained in three sets of experiments and assessed in triplicate for method validation.

To ascertain if both methods were equally valid in measuring the antioxidant activity, they were correlated through the Pearson coefficient index (−1 < *r* < 1). A Pearson coefficient absolute value higher than 0.9 shows a strong correlation between the two methods. The Pearson index was also used to correlate the data of antioxidant activity with the quantification of the several chemical classes of compounds to ascertain their relationship with antioxidant power. Once both *Asphodelus* species were collected in two different seasons, the obtained data were also analyzed by ANOVA, with the season as the source of variance. The significance value was set for a *p*-value < 0.05.

### 3.8. In Vitro Genotoxicity/Mutagenicity Evaluation by Ames Test

A bacterial reverse mutation test (Ames test), commonly employed as an initial screening of the genotoxicity potential of herbal substances/preparations, was used to detect relevant genetic changes and genotoxic carcinogens [54]. The assessment of mutagenicity was performed according to the OECD No. 471 [55], the ICH S2 (R1) [56] guidelines, and following the published protocols [44], using five *Salmonella enterica* serovar Typhimurium tester strains (TA98, TA100, TA102, TA1535, and TA1537) in a direct plate incorporation method with and without metabolic activation. TA100, TA98, TA102 and TA1535 were kindly provided by the Genetic Department of the Nova Medical School of the Universidade NOVA de Lisboa (Portugal), having received them from Professor B.N. Ames (Berkeley, CA, USA). TA1537 was from ATCC, NUMBER: 29630™, LOT: 7405375. The strains were inoculated in nutrient broth medium and incubated for 12–16 h, at 37 °C in the dark, shaking at 210 rpm in an orbital incubator, and kept at 4 °C until use.

S9 mix (10%, *v/v* rat liver S9, 0.4 M MgCl_2_, 1.65 M KCl, 1 M glucose-6-phosphate, 0.1 M nicotinamide adenine dinucleotide phosphate, and 0.2 M sodium phosphate buffer, pH 7.4) was freshly prepared and kept on ice during the experiment.

The extracts (25 mg/mL) were dissolved in DMSO (up to 30%), which also served as the negative control. An amount of 200 µL of extract dilutions were mixed with 500 µL sodium phosphate buffer (0.1 M, pH 7.4) (assay without metabolic activation) or S9 mix (assay with metabolic activation), 100 µL of the bacterial culture, and 2 mL of melted top-agar, supplemented with 0.05 mM biotin and histidine, at 45 °C. This mixture was then vortexed and plated on Petri dishes with Vogel-Bonner agar medium and supplemented with 2% glucose. After a 48-h incubation at 37 °C, manual counting of His+ revertant colonies for each concentration was performed. All assays were performed in triplicate. The results were expressed as the mean number of revertant colonies with the standard deviation (mean ± SD). The positive controls were sodium azide (SA, 1.5 µg/plate for TA100 and TA1535), 2-nitrofluorene (2-NF, 5 µg/plate for TA98), 9-aminoacridine (9-AA, 100 µg/plate for TA1537), and tert-butyl hydroperoxide (tBHP, 50 µg/plate for TA102) in the assay without metabolic activation, and 2-aminoathracene (2-AA, 2 µg/plate for TA98 and 10 µg/plate for TA102, TA1535 and TA1537) and benzo(*a*)pyrene (BaP, 5 µg/plate for TA100) in the assay with metabolic activation.

### 3.9. In Vitro Cytotoxicity Evaluation by MTT Assay

Cytotoxicity was evaluated by the methylthiazolyldiphenyl-tetrazolium bromide (MTT) reduction assay [57] on a human liver cell line HepG2 (ATCC Cat. No. HB-8065, Middlesex, UK). HepG2 were seeded in 96-well plates at a density of 8.5 × 10^4^ cells/cm^2^ in α-MEM (Sigma-Aldrich^®^, St. Louis, MO, USA) with 1 mM sodium pyruvate (PAN Biotech, Aidenbach, Germany) and 1% non-essential amino acids (NEAA, PAN Biotech, Aidenbach, Germany) supplemented with 10% fetal bovine serum (FBS, Gibco^®^- Thermo Fisher Scientific^TM^ (Waltham, MA, USA), in a humidified chamber at 37 °C in a 5% CO_2_ atmosphere. After 48-h incubation, the cell culture medium was replaced by fresh medium with AbLa and AmLa extracts (9:1) at final concentrations of 50, 125, 250, 500, and 1000 µg/mL. Cells were also incubated with a complete cell culture medium, DMSO 1% and DMSO 20% in α-MEM as a positive, solvent, and negative control, respectively. After 48 h, the cells were carefully washed with 100 μL PBS, and 200 μL 0.5 mg/mL MTT (Sigma- Aldrich^®^) in a cell culture medium was added. HepG2 were incubated for 3 h in a humidified chamber at 37 °C in a 5% CO_2_ atmosphere. The purple crystals were solubilized with 200 μL DMSO and measured at 570 nm using a microplate spectrophotometer (SPECTROstar Omega; BMG LabTech, Ortengerg, Germany). The results were expressed as a percentage relative to the solvent control. Four wells were used for each sample, and at least two independent experiments were performed.

Data analysis and graphs were plotted using GraphPad Prism^®^ software (version 9.0.0.121, GraphPad Software, San Diego, CA, USA). Results are presented as mean ± standard deviation. *p* < 0.05 was considered significant.

## 4. Conclusions

The weak antimicrobial activity verified with our crude leaf extracts of *Asphodelus bento-rainhae* and *Asphodelus macrocarpus* is consistent with the results obtained when testing other *Asphodelus* spp. against a similar panel of pathogens [4]. However, fractionation of these extracts enabled the detection of significant antimicrobial activity in the diethyl ether L-L partition fractions, quantitatively the richest in 1,8-dihydroxyanthracene derivatives, a known chemical class of secondary metabolites with antimicrobial activity [34]. Furthermore, the well-known antibacterial agent aloe-emodin was identified as the main compound responsible for this activity. Although the in vitro cytotoxicity and mutagenicity of this compound has been reported by others, no cytotoxicity or mutagenic activity was observed in the corresponding extracts and fractions that we tested.

On the other hand, the ethyl acetate L-L partition fractions are quantitatively the richest in phenolic acids and flavonoid derivatives, and showed the highest antioxidant activity, confirming the major role of the different classes of the identified phenolic compounds in the activity of *Asphodelus bento-rainhae* and *Asphodelus macrocarpus* leaves as medicinal plants. Moreover, the negative results of the Ames and MTT tests indicate that the hydroethanolic leaf extracts of both species are safe in terms of toxicity, and these data together with the phytochemical profiles will provide appropriate information for inclusion in the future quality monograph of these medicinal plants.

## Figures and Tables

**Figure 1 molecules-28-02372-f001:**
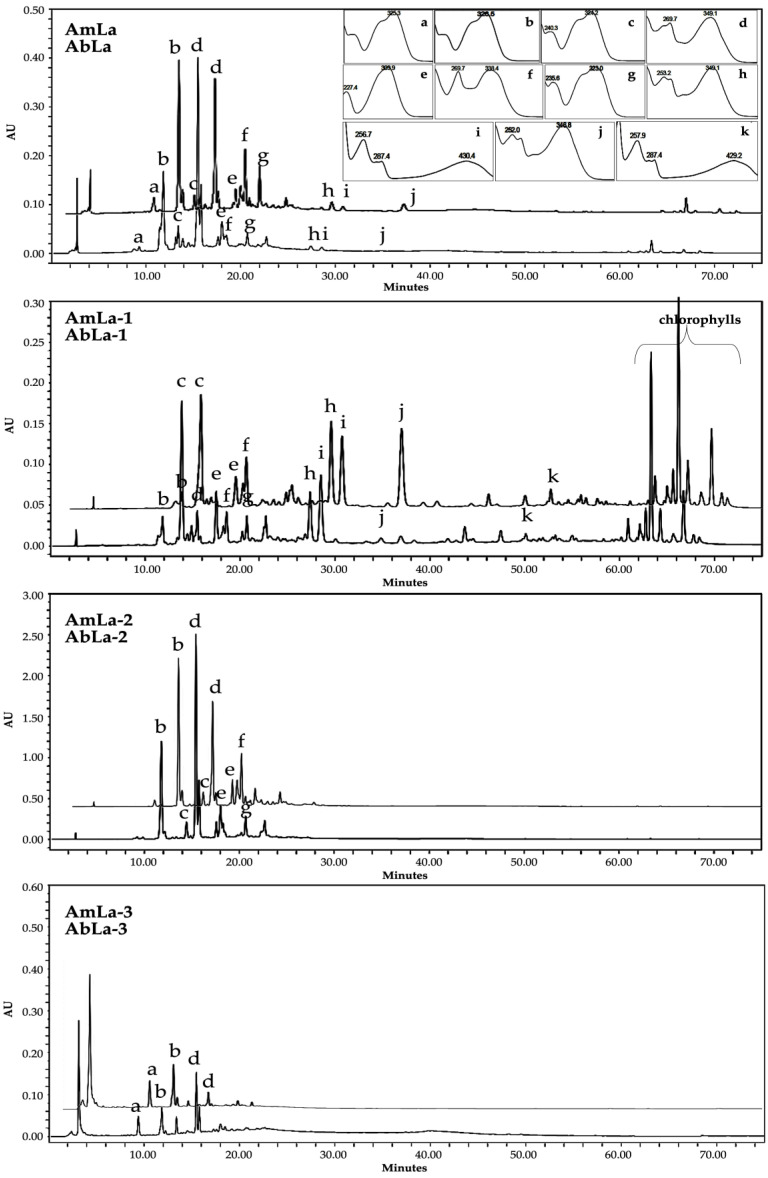
Comparative HPLC-UV/DAD chromatographic profiles of marker secondary metabolites of *A. bento-rainhae* and *A. macrocarpus* leaf crude extracts and their subsequent L-L partitions. Abbreviations: AbLa: *A. bento-rainhae* leaf first collection, AmLa: *A. macrocarpus* leaf first collection, (−1): ethyl ether fractions, (−2): ethyl acetate fractions, and (−3): aqueous fractions.

**Figure 2 molecules-28-02372-f002:**
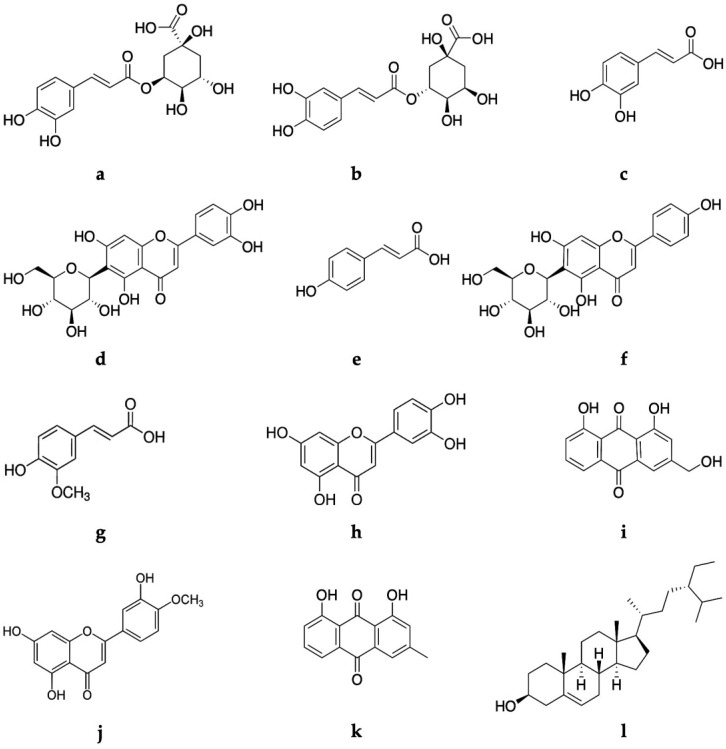
Structures of the marker secondary metabolites (a to l) from *A. bento-rainhae* and *A. macrocarpus* leaf extracts mentioned in Table 1.

**Figure 3 molecules-28-02372-f003:**
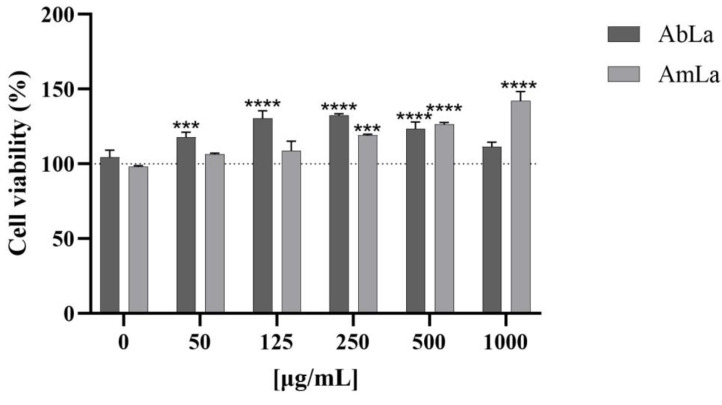
HepG2 viability after 48 h of incubation with AbLa and AmLa extracts evaluated by MTT reduction assay. Data are shown as the percentage of solvent control (dashed line) and as mean ± standard deviation; *n* = 2–5. *** *p* < 0.001; **** *p* < 0.0001.

**Table 1 molecules-28-02372-t001:** Characterization of the peaks of interest obtained from *A. bento-rainhae* and *A. macrocarpus* leaf crude extracts and their subsequent L-L partitions.

Peak	t_R_ (min)	λ_max_ (nm)	[M-H]^−^(*m/z*)	MS/MS (*m/z*)	Identified Compound
a	9.29	325.3	353	191 (100), 179 (3)	* neochlorogenic acid
b	11.79	326.5	353	191 (100), 179 (67)	chlorogenic acid
c	13.82	240.3, 324.2	179	135 (100)	caffeic acid
d	15.43	269.7, 349.1	447	357 (43), 327 (100), 297 (76)	isoorientin
e	17.46	227.4, 309.9	163	119 (100)	*p*-coumaric acid
f	18.55	269.7, 338.4	431	341 (23), 311 (72), 283 (100)	isovitexin
g	20.53	235.6, 323.0	193	178 (62), 149 (68), 134 (100)	ferulic acid
h	27.38	253.2, 349.1	285	175 (13), 151 (100), 133 (22)	luteolin
i	28.73	256.7, 287.4, 430.4	269	239 (100)	aloe-emodin
j	34.78	252.0, 346.8	299	284 (100)	diosmetin
k	50.46	257.9, 287.4, 429.2	253	225 (100)	chrysophanol

Abbreviations: t_R_: Retention time, λ_max_: wavelength. * neochlorogenic acid is a synonym name of 5-*O*-caffeoylquinic acid.

**Table 2 molecules-28-02372-t002:** Quantification of the principal classes of secondary metabolites of *A. bento-rainhae* and *A. macrocarpus* leaf crude extracts.

Assays	AbLa	AbLb	AmLa	AmLb
	Mean ± SD	Mean ± SD	Mean ± SD	Mean ± SD
**TPC** (mg GAE/g dried extract) (mg GAE/g dried leaf)	44.16 ± 21.62 9.23 ± 4.52	38.83 ± 17.1 8.57 ± 3.78	37.15 ± 14.32 12.63 ± 5.38	38.28 ± 15.63 6.09 ± 2.49
**TFC** (mg CAE/g dried extract) (mg CAE/g dried leaf)	* 40.79 ± 4.45 8.16 ± 0.89	29.56 ± 1.43 6.53 ± 0.32	33.46 ± 0.89 5.32 ± 0.14	* 35.52 ± 1.51 12.08 ± 0.51
**TAC** (mg RhE/g dried extract) (mg RhE/g dried leaf)	* 1.16 ± 0.13 0.24 ± 0.05	1.07 ± 0.11 0.24 ± 0.04	0.55 ± 0.07 0.19 ± 0.02	0.81 ± 0.09 0.13 ± 0.01
**TCTC** (mg CAE/g dried extract) (mg CAE/g dried leaf)	* 180.96 ± 10.98 37.82 ± 2.30	149.71 ± 12.98 33.06 ± 2.87	132.60 ± 2.73 45.09 ± 0.93	142.98 ± 6.71 22.73 ± 1.07
**THTC** (mg GAE/g dried extract) (mg GAE/g dried leaf)	67.61 ± 9.22 14.13 ± 1.93	* 55.16 ± 6.64 12.18 ± 1.47	60.53 ± 8.04 20.58 ± 2.74	37.03 ± 3.87 5.89 ± 0.62
**TTC** (mg OAE/g dried extract) (mg OAE/g dried leaf)	111.72 ± 22.77 23.35 ± 4.76	88.78 ± 23.22 19.60 ± 5.13	* 165.47 ± 26.54 56.26 ± 9.03	125.74 ± 20.72 19.99 ± 3.29

Abbreviations: AbLa: *A. bento-rainhae* leaf first collection, AbLb: *A. bento-rainhae* leaf second collection, AmLa: *A. macrocarpus* leaf first collection, AmLb: *A. macrocarpus* leaf second collection, TPC: total phenolic content, TFC: total flavonoid content, TAC: total anthraquinones content, TCTC: total condensed tannin content, THTC: total hydrolysable tannin content, TTC: total triterpenoid content, GAE: gallic acid equivalents, CAE: catechin equivalents, RhE: rhein equivalents, OAE: oleanolic acid equivalents. * Significantly higher content (*p*-value < 0.05) when compared between different species of the same collection season analyzed by ANOVA test.

**Table 3 molecules-28-02372-t003:** In vitro determination of the antioxidant activity of *A. bento-rainhae* and *A. macrocarpus* leaf crude extracts and their subsequent L-L partitions.

Extracts Code	Assays	
	DPPH (IC_50_ μg/mL)	FRAP (mmol AA/g Dry Extract)
AbLa	2000	0.337 ± 0.042
AbLb	2540	0.306 ± 0.023
AmLa	2990	0.280 ± 0.046
AmLb	3070	0.271 ± 0.072
AbLa-1	2950	Nd
AbLa-2	800	Nd
AbLa-3	2910	Nd
AmLa-1	3009	Nd
AmLa-2	1200	Nd
AmLa-3	4000	Nd
AA	83	Nd

Abbreviations: AbLa: *A. bento-rainhae* leaf first collection extract, AbLb: *A. bento-rainhae* leaf second collection extract, AmLa: *A. macrocarpus* leaf first collection extract, AmLb: *A. macrocarpus* leaf second collection extract, DPPH: 2,2-diphenyl-1-picrylhydrazyl, IC_50_: The half maximal inhibitory concentration, FRAP: Ferric reducing antioxidant power, AA: ascorbic acid, Nd: not determined.

**Table 4 molecules-28-02372-t004:** In vitro antimicrobial activity of *A. bento-rainhae* and *A. macrocarpus* leaf etheric L-L partition extracts against Gram-positive strains.

Bacteria (Gram +)	MIC (µg/mL)		
	AbLa-1	AmLa-1	Aloe-Emodin
*S. aureus* ATCC 29213	500	500	3.2
*S. aureus* CQINSA4923	62	125	50
*S. aureus* INSArefV	500	500	1.6
*S. aureus* INSA936	250	250	12.5
*S. aureus* INSA896	125	125	3.2
*S. saprophyticus* INSA842	125	250	100
*S. saprophyticus* INSA867	1000	1000	25
*S. epidermidis* INSA796	250	500	1.6
*S. epidermidis* INSA958	250	500	0.8
*S. epidermidis* INSA960	125	125	1.6
*S. haemolyticus* INSA982	125	125	25
*S. haemolyticus* INSA984	125	125	12.5

Abbreviations: AbLa: *A. bento-rainhae* leaf first collection extract, AmLa: *A. macrocarpus* leaf first collection extract, ATCC: American Type Culture Collection, INSA, Instituto Nacional de Saúde clinical strains collection, MIC: minimum inhibitory concentration.

**Table 5 molecules-28-02372-t005:** Mutagenicity of *A. bento-rainhae* and *A. macrocarpus* leaf crude extracts in the bacterial reverse mutation test (Ames Test).

AbLa µg/Plate	Number of Revertant Colonies Without Metabolic Activation, Mean (*n* = 3) ± Standard Deviation (SD)				
	TA98	TA100	TA102	TA1535	TA1537
250	17 ± 4	160 ± 7	355 ± 13	19 ± 4	10 ± 1
625	20 ± 4	158 ± 5	349 ± 34	24 ± 1	10 ± 2
1250	17 ± 2	182 ± 16	429 ± 25	20 ± 1	7 ± 1
2500	21 ± 2	178 ± 8	458 ± 16	22 ± 2	8 ± 2
3750	24 ± 3	175 ± 19	472 ± 29	21 ± 4	9 ± 3
5000	24 ± 2	175 ± 14	485 ± 31	18 ± 1	13 ± 3
**AmLa** µg/plate					
250	17 ± 2	186 ± 10	357 ± 14	22 ± 3	9 ± 1
625	20 ± 2	155 ± 15	365 ± 3	20 ± 3	9 ± 2
1250	22 ± 5	150 ± 5	394 ± 8	16 ± 1	10 ± 5
2500	21 ± 3	170 ± 15	441 ± 2	17 ± 3	12 ± 5
3750	24 ± 5	168 ± 4	454 ± 24	17 ± 3	8 ± 2
5000	23 ± 3	165 ± 20	407 ± 28	24 ± 2	15 ± 1
NC	19 ± 2	156 ± 17	320 ± 4	21 ± 3	7 ± 1
PC	2-NF	SA	tBHP	SA	9-AA
	488 ± 30	1048 ± 43	881 ± 26	827 ± 13	1354 ± 5
**AbLa** µg/plate	**Number of revertant colonies with metabolic activation, mean (*n* = 3) ± standard deviation (SD)**				
500	Nd	166 ± 22	221 ± 16	19 ± 4	15 ± 1
1250	63 ± 6	164 ± 9	248 ± 11	15 ± 7	16 ± 1
2500	59 ± 5	174 ± 4	248 ± 11	17 ± 2	11 ± 1
5000	52 ± 6	178 ± 15	254 ± 12	15 ± 1	16 ± 1
NC	44 ± 8	157 ± 6	172 ± 2	11 ± 2	12 ± 1
PC	2-AA	BaP	2-AA	2-AA	2-AA
	832 ± 35	947 ± 148	732 ± 12	266 ± 1	306 ± 50

Abbreviations: AbLa: *A. bento-rainhae* leaf first collection extract, AmLa: *A. macrocarpus* leaf first collection extract, Nd: not determined, NC: negative control/solvent control (DMSO 30%), PC: positive control reference, 2-NF: 2-nitrofluorene, SA: sodium azide, tBHP: tert-butyl hydroperoxide, 9-AA: 9-aminoacridine, 2-AA: 2-aminoathracene, BaP: benzo(*a*)pyrene.

**Table 6 molecules-28-02372-t006:** Composition of the Gram-positive pathogen panel under study.

Bacteria (Gram +)		Demonstration of Resistance to the Antibiotics										
	CXT	CPFX	DAP	ERY	FA	GN	Lzd	OXA	PCN	TEC	TET	VAN
*S. aureus* ATCC 29,213								S	MS			
*S. aureus* CQINSA4923	R	R		R	S	R	S	R	R	S	S	S
*S. aureus* INSArefV	R									R		R
*S. aureus* INSA936			R									
*S. aureus* INSA896	R	R			R		R					
*S. saprophyticus* INSA842				R	R							
*S. saprophyticus* INSA867										R		
*S. epidermidis* INSA796	R	R					R			R		
*S. epidermidis* INSA958							R			R		
*S. epidermidis* INSA960										R		
*S. haemolyticus* INSA982	R	R								R		
*S. haemolyticus* INSA984	R	R	R									

Abbreviations: ATCC: American Type Culture Collection, INSA: Instituto Nacional de Saúde clinical strains collection, CXT: cefoxitin, CPFX: ciprofloxacin, DAP: daptomycin, ERY: erythromycin, FA: fusidic acid, GEN: gentamicin, Lzd: linezolid, OXA: oxacillin, PCN: penicillin, TEC: teicoplanin, TET: tetracycline, Van: vancomycin, MR: methicillin-sensitive, S: sensitive, R: resistant.

## Data Availability

Not applicable.

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
