# Peer review of "Identification of Marker Compounds and In Vitro Toxicity Evaluation of Two Portuguese Asphodelus Leaf Extracts"

_molecules, 2023, doi:10.3390/molecules28052372_

Round 1
Reviewer 1 Report
The article can be accepted after minor revision. The corresponding author can find my remarks or suggestions directly on the article.
Proofreading by a native English speaker should be necessary to improve the writing.

Author Response
Response to Reviewer 1 Comments
Reviewer comments:
The article can be accepted after minor revision. The corresponding author can find my remarks or suggestions directly in the article.
Authors response: Dear reviewer, thank you very much for your comments. We have attentively addressed all the comments in the revised version of the manuscript as follows:
Point 1. In the lines 18 to 20, Please rewrite this sentence which is not very clear. What does "their hydroethanolic extracts" means? Hydroethanolic extracts of leaves of the studied plants I suppose. for example, it may be "The aim of this study is to establish the phytochemical profile of the studied plants by highlighting their main marker compounds in the leaves and to evaluate their antimicrobial and antioxidant activities, and also the toxicity of their hydroethanolic extracts.
Authors response 1: To clarify the objectives of the study, the following text was modified.
“The present study aims to establish the phytochemical profile of the main secondary metabolites together with the antimicrobial, antioxidant and toxicity assessments of both Asphodelus leaf 70% ethanol extracts”.
Point 2. In the lines 23 to 25, better to rewrite: "For in vitro evaluations of antimicrobial activity, broth microdilution method, and for the antioxidant activity, FRAP and DPPH methods were used."
Authors response 2: Dear reviewer, we agree, and the following text was modified accordingly:
“For in vitro evaluations of antimicrobial activity, broth microdilution method, and for the antioxidant activity, FRAP and DPPH methods were used.”
Point 3. In the line 27, ρ-coumaric acid, should be replaced by p-coumaric acid (p in italic).
Authors response 3: p-coumaric acid was correctly inserted in the manuscript.
Point 4. In the lines 28 to 29, “Terpenoids (111.72±22.77 and 165.47±26.54 mg OAE/ g dry weight) and condensed tannins (180.96±10.98 and 132.60±2.73 mg CAE/ g dry weight)”, it is not very useful to give these values in the abstract.
Authors response 4: The referred values were removed from the abstract.
Point 5. In the line 46, “Asphodelus L. species”, put the name of the family in brackets after species.
Authors response 5: The family name was placed in the text accordingly.
“Asphodelus L. species (Asphodelaceae) are consumed…...”
Point 6. In the line 49, “ciris otu”, the correct name is "çiriÅŸ otu"
Authors response 6: The correct name "çiriÅŸ otu" was placed in the text accordingly.
Point 7. In the lines 57 to 60, “Several in vitro and in vivo biological activities of Asphodelus spp. leaf and aerial parts extracts have been reported”, Roots also. I think the authors may say "leaf, aerial parts and root extracts..." even they only deal with leaves in this article.
Authors response 7: Dear reviewer, the authors would like to keep the following sentences as the mentioned bibliographic references specifically referred to the phytochemical and biological activities of leaves and aerial parts as plant material, the same plant parts in the experimental work we reported in this manuscript.
Point 8. In vitro and in vivo in italic.
Authors response 8: Dear reviewer, following the MDPI journals style, terms like “etc.” (or et cetera when spelled in full) “et al.” and “in vitro” / “in vivo” / “ex vivo,” don’t need to be in italics and therefore are not italicized in the manuscript.
Point 9. In the line 61, according to the world flora online and the world checklist of vascular plants (WCVP), Asphodelus bento-rainhae has two subspecies, A. bento-rainhae subsp. bento-rainhae and A. bento-rainhae. subsp. salmanticus. Please check again what do you mean by writing only one of these subspecies in square brackets.
Authors response 9: Dear reviewer, although there are two subspecies of Asphodelus bento-rainhae worldwide, only one of them “Asphodelus bento-rainhae subsp. bento-rainhae” is an endemic species of Portugal and is the subject of this study.
Point 10. In Figure 1, since a and a', b and b' etc. are the same compounds, you have to put only one letter (a, b etc.) on the figure. Even if the compounds are found in two different plants, their structures are the same.
Authors response 10: Dear reviewer, Figure 1 was modified accordingly, and the same letters are given to the same peaks and structures of the main marker compounds.
Point 11. In the lines 110 to 115, Some plants' names are complete with the botanists' names, some others not. Please harmonize the writing and indicate the whole name for each plant cited for the first time in the text.
Authors response 11: Dear reviewer, we absolutely agree with your comment, the scientific name of each species cited for the first time, is completely spelled out in the manuscript (e.g., Asphodelus bento-rainhae subsp. bento-rainhae P. Silva). Then for the second time, the common scientific name (e.g., Asphodelus bento-rainhae) and later on, an abbreviation of the genus (e.g., A. bento-rainhae) is mentioned.
Point 12. In Figure 2,
-put only one letter under the structures: a, b or c etc. since a and a' are the same and so on.
-correct the structure of g: OHC3 must be OCH3
-since the h (luteolin) is the aglycone of the d (isoorientin), draw the same way the phenyl moiety.
-illustrate the carbons bearing methoxy group on g and j the same way
Authors response 12: Dear reviewer, all the referred modifications were applied to Figure 2.
Point 13. In the line 134, -put this information “(p-value: 0.01)” at the end of the sentence-writing all p in italic.
Authors response 13: Dear reviewer, referred corrections were addressed in the revised version of the manuscript.
Point 14. In the lines 182 to 183, Better write "the data ... correlate" since data is plural here.
Authors response 14: Dear reviewer, referred corrections were addressed in the revised version of the manuscript.
Point 15. In figure 3, the resolution of this graph is not of good quality.
Authors response 15: Dear reviewer, Figure 3 was replaced with a higher-resolution graph.
Moreover, several other grammatical issues such as spaces, parenthesis, commas (,), and two points (:) have been corrected throughout the manuscript.
Reviewer 2 Report
The manuscript in reference describes a study oriented to establish the main phytochemical-based markers, including antimicrobial, antioxidant, and toxicity effects of the hydroethanolic extracts of two Portugal-grown Asphodelus plants. The manuscript is interesting and includes relevant information for readers. However, some points should be addressed before being considered further.
1. Detailed scrutiny should be performed throughout the manuscript to look for some grammar, stylistic, and even typos issues.
2. Line 20: Add ethanol after 70%, i.e., 70% ethanol
3. Lines 47-68: Rewrite the passages in these paragraphs since a lack of connection is evidenced, i.e., there are short paragraphs with low connectivity.
4. Figure 1: Improve the quality and resolution of this figure.
5. Table 1: Add the MS fragments obtained after MS analysis using a triple quadrupole analyzer. In addition, [M-H]- is not a molecular ion but a quasimolecular ion. Be consistent throughout the manuscript.
6. There is a lack of sufficient discussion comparing the results with the reported studies.
7. Line 95: An adequate description of MS analysis leading to compound identification is missing. The authors must provide pertinent details about how experimental MS information was used to provide compound identification, despite being known compounds. Without this, the manuscript lacks scientific quality.
8. Line 170: Include the test to define significant differences.
9. Line 213: Extracts exhibited poor antibacterial activity. This fact must clearly be described and discussed in the manuscript.
10. Table 3: Revise if the IC50 for DPPH assay is expressed as mg/mL since it is related to very high concentrations (in fact, it should be in µg/mL) and seems to be incorrect.
11. Conclusions must be improved since it is very global and should be improved from a mechanistic point of view.
Author Response
Response to Reviewer 2 Comments
Reviewer comments:
" The manuscript in reference describes a study oriented to establish the main phytochemical-based markers, including antimicrobial, antioxidant, and toxicity effects of the hydroethanolic extracts of two Portugal-grown Asphodelus plants. The manuscript is interesting and includes relevant information for readers. However, some points should be addressed before being considered further."
Authors response: Dear reviewer, thank you very much for your comments. We have attentively addressed all the comments in the revised version of the manuscript as follows:
Point 1: Detailed scrutiny should be performed throughout the manuscript to look for some grammar, stylistic, and even typos issues.
Authors response: Dear reviewer, according to your comment, a detailed scrutiny was performed throughout the manuscript.
Point 2: Line 20: Add ethanol after 70%, i.e., 70% ethanol.
Authors response: 70% ethanol was placed in the manuscript.
Point 3: Lines 47-68: Rewrite the passages in these paragraphs since a lack of connection is evidenced, i.e., there are short paragraphs with low connectivity.
Authors response: Dear reviewer, the referred lines and paragraphs were modified accordingly as followed:
“Asphodelus L. species (Asphodelaceae) are consumed in large quantities in the cuisine (e.g., soups, pastries, etc.) of several countries and different cultures. Leaves of Asphodelus aestivus Brot., for instance, are commonly consumed as a cooked vegetable dish in Turkey, where is known as “çiriÅŸ otu” [2]. In Puglia, on the southeast coast of Italy, burrata cheese is always wrapped in Asphodelus ramosus L. leaves to indicate the freshness of the cheese until dried out [3]. In addition to the nutritional values, Asphodelus spp. leaves are widely used in traditional medicine to treat ulcers, urinary and inflammatory disorders [4]. In North African countries and the Iberian Peninsula, decoctions of leaves and stems have also been used to treat withering and paralysis [5,6]. Previously reported phytochemical studies of Asphodelus spp. extracts from leaves and aerial parts revealed the presence of phenolic acids [7,8], flavonoids [6–11] and anthracene derivatives [8,12–16] as the main chemical classes of their marker secondary metabolites. Several in vitro and in vivo biological activities of Asphodelus spp. leaf and aerial parts extracts have been reported and documented for antimicrobial [7,15,17–20], antioxidant [2,18,20–22], and antitumoral [7,15,21,24] activities [4].
Asphodelus bento-rainhae subsp. bento-rainhae P. Silva, is an endemic species from Serra da Gardunha, considered “Vulnerable” on the Red List of Threatened Species of the International Union for the Conservation of Nature (IUCN) and is co-existing with Asphodelus macrocarpus subsp. macrocarpus Parl. in the same geographical area. They are known by the common Portuguese name “abrotea” (Ancient Greek: ἈβρÏŒτονον) and their leaf is used as fertilizer and fodder in Portugal [25]. Up to this date, no data related to the phytochemical characterization, pre-clinical safety and biological potential of Asphodelus bento-rainhae and Asphodelus macrocarpus leaves were found in the literature. Therefore, the present study was conducted to identify the main chemical constituents, antimicrobial and antioxidant activities of leaf extracts of these species along with their in vitro toxicity assessments, using samples collected at different times of the year to determine the most appropriate period for the collection of material and to contribute to the knowledge of safety and their value as herbal medicinal products”.
Point 4: Figure 1: Improve the quality and resolution of this figure.
Authors response: Figure 1 with higher resolution was placed in the manuscript.
Point 5: Table 1: Add the MS fragments obtained after MS analysis using a triple quadrupole analyzer. In addition, [M-H]- is not a molecular ion but a quasimolecular ion. Be consistent throughout the manuscript.
Authors response: Dear reviewer, MS fragments were added to the Table 1 and [M-H]− (m/z) was modified accordingly.
Point 6: There is a lack of sufficient discussion comparing the results with the reported studies.
Authors response: The reviewer´s comment have been addressed throughout the manuscript.
Point 7: Line 95: An adequate description of MS analysis leading to compound identification is missing. The authors must provide pertinent details about how experimental MS information was used to provide compound identification, despite being known compounds. Without this, the manuscript lacks scientific quality.
Authors response: Dear reviewer, the identification of the compounds was performed based on both TLC and HPLC-UV spectral characteristics, using co-chromatography with purchased standard compounds and with comparison with literature data. Therefore, considering the obtained characteristics with exact match to the reference standards and to keep the consistency of the text, these known details are not described in the manuscript. However, the missing information of the techniques and standards were modified.
- In the section "2.2. Phytochemical analysis" (Page 2, line 91) of "Results and Discussion":
“…. spectral analysis, using standards co-chromatography and comparison with literature data."
- In the section "3.1 chemical and biological reagents" (Page 11, lines 296, 305 to 307 and 310 to 311) of "Materials and Methods”:
“…. benzo(a)pyrene, chlorogenic acid, chrysophanol, d-(+)-biotin,…”
“Aloe-emodin, caffeic acid, (+)-catechin, ferulic acid, isoorientin, isovitexin, luteolin, oleanolic acid, p-coumaric acid and rhein were acquired from Extrasynthese (Genay, France).”
“…. β-sitosterol and 2-aminoethyl diphenylborinate were from Acros organics (Geel, Belgium).”
- In the section "3.4. chromatographic conditions" (Page 12, lines 367 to 369) of " Materials and Methods:
“Peaks assignment and the identification of compounds were based on a co-chromatography technique with comparison of retention times, UV-DAD and mass spectral data with those of standards and published data.”
Point 8: Line 170: Include the test to define significant differences.
Authors response: the correspondent test to define the significant differences was added.
Point 9: Line 213: Extracts exhibited poor antibacterial activity. This fact must clearly be described and discussed in the manuscript.
Authors response: Dear reviewer, the activity in question was modified in the revised version.
- In the section "2.4. Assessment of the antibacterial potential" (Page 6, lines 251 to 256) of "Results and Discussion":
“Overall, the observed antimicrobial activity of both A. bento-rainhae and A. macrocarpus leaf crude extracts were similar to those obtained and reported form the other Asphodelus spp. tested against a similar panel of pathogens. However, fractionation of crude extracts enabled the detection of significant antimicrobial activity in the diethyl ether L-L partition fractions, quantitatively the richest in 1,8-dihydroxyanthracene derivatives, a known chemical class of secondary metabolites with antimicrobial activity [34].”
- In the “Conclusions” (lines 507 to 524):
“The weak antimicrobial activity verified with our crude leaf extracts of Asphodelus bento-rainhae and Asphodelus macrocarpus is consistent with results obtained, when testing other Asphodelus spp. against a similar panel of pathogens [4]. However, fractionation of these extracts enabled the detection of significant antimicrobial activity in the diethyl ether L-L partition fractions, quantitatively richest in 1,8-dihydroxyanthracene derivatives, a known chemical class of secondary metabolites with antimicrobial activity [34]. Furthermore, the well-known antibacterial agent aloe-emodin was identified as the main compound responsible for this activity.”
Point 10: Table 3: Revise if the IC50 for DPPH assay is expressed as mg/mL since it is related to very high concentrations (in fact, it should be in µg/mL) and seems to be incorrect.
Authors response: values of the IC50 were corrected in the revised version of the manuscript.
Point 11: Conclusions must be improved since it is very global and should be improved from a mechanistic point of view.
Authors response: Dear reviewer, the conclusions (lines 507 to 524) were improved accordingly.
“The weak antimicrobial activity verified with our crude leaf extracts of Asphodelus bento-rainhae and Asphodelus macrocarpus is consistent with results obtained, when testing other Asphodelus spp. against a similar panel of pathogens [4]. However, fractionation of these extracts enabled the detection of significant antimicrobial activity in the diethyl ether L-L partition fractions, quantitatively richest in 1,8-dihydroxyanthracene derivatives, a known chemical class of secondary metabolites with antimicrobial activity [34]. Furthermore, the well-known antibacterial agent aloe-emodin was identified as the main compound responsible for this activity. Although the in vitro cytotoxicity and mutagenicity of this compound has been reported by others, no cytotoxicity or mutagenic activity was observed in the corresponding extracts and fractions that we tested.
On the other hand, the ethyl acetate L-L partition fractions are quantitatively the richest in phenolic acids and flavonoid derivatives and showed the highest antioxidant activity, confirming the major role of the different classes of the identified phenolic compounds in the activity of Asphodelus bento-rainhae and Asphodelus macrocarpus leaves as medicinal plants. Moreover, the negative results of the Ames and MTT tests indicate that the hydroethanolic leaf extracts of both species are safe in terms of toxicity and these data together with the phytochemical profiles will provide appropriate information for inclusion in the future quality monograph of these medicinal plants.”
Round 2
Reviewer 2 Report
The authors addressed my comments adequately so the manuscript can be accepted in its current form.